# Algae-Derived Natural Products in Diabetes and Its Complications—Current Advances and Future Prospects

**DOI:** 10.3390/life13091831

**Published:** 2023-08-29

**Authors:** Leonel Pereira, Ana Valado

**Affiliations:** 1Department of Life Sciences, University of Coimbra, Calçada Martim de Freitas, 3000-456 Coimbra, Portugal; 2MARE-Marine and Environmental Sciences Centre/ARNET-Aquatic Research Network, University of Coimbra, 3000-456 Coimbra, Portugal; valado@estesc.ipc.pt; 3Biomedical Laboratory Sciences, Polytechnic Institute of Coimbra, Coimbra Health School, Rua 5 de Outubro-SM Bispo, Apartado 7006, 3045-043 Coimbra, Portugal

**Keywords:** diabetes, natural products, algae, antidiabetic, glycemic control, therapeutic potential

## Abstract

Diabetes poses a significant global health challenge, necessitating innovative therapeutic strategies. Natural products and their derivatives have emerged as promising candidates for diabetes management due to their diverse compositions and pharmacological effects. Algae, in particular, have garnered attention for their potential as a source of bioactive compounds with anti-diabetic properties. This review offers a comprehensive overview of algae-derived natural products for diabetes management, highlighting recent developments and future prospects. It underscores the pivotal role of natural products in diabetes care and delves into the diversity of algae, their bioactive constituents, and underlying mechanisms of efficacy. Noteworthy algal derivatives with substantial potential are briefly elucidated, along with their specific contributions to addressing distinct aspects of diabetes. The challenges and limitations inherent in utilizing algae for therapeutic interventions are examined, accompanied by strategic recommendations for optimizing their effectiveness. By addressing these considerations, this review aims to chart a course for future research in refining algae-based approaches. Leveraging the multifaceted pharmacological activities and chemical components of algae holds significant promise in the pursuit of novel antidiabetic treatments. Through continued research and the fine-tuning of algae-based interventions, the global diabetes burden could be mitigated, ultimately leading to enhanced patient outcomes.

## 1. Introduction

### 1.1. Overview of Diabetes and Its Complications

Diabetes, a chronic metabolic disorder characterized by impaired insulin production or utilization, has reached epidemic proportions worldwide. The disease affects millions of individuals, imposing a significant burden on healthcare systems and economies. Diabetes is associated with various complications that significantly impact the quality of life and increase morbidity and mortality rates among affected individuals [1,2].

Type 2 diabetes mellitus (T2DM) is the most prevalent form, accounting for approximately 90% of all diabetes cases [3]. It arises due to a combination of insulin resistance and inadequate insulin secretion. T2DM is often associated with lifestyle factors such as sedentary behavior, unhealthy dietary habits, and obesity [4]. Conversely, type 1 diabetes (T1DM) is an autoimmune condition characterized by the destruction of insulin-producing beta cells in the pancreas, resulting in absolute insulin deficiency [5]. Other forms of diabetes, such as gestational diabetes, also pose significant health risks, particularly to pregnant women and their offspring [6].

The complications associated with diabetes are diverse and affect multiple organ systems. Chronic hyperglycemia in diabetes leads to the development of microvascular and macrovascular complications. Microvascular complications primarily affect the small blood vessels, including the eyes (diabetic retinopathy), kidneys (diabetic nephropathy), and nerves (diabetic neuropathy). Macrovascular complications involve the body’s large blood vessels and contribute to an increased risk of cardiovascular diseases such as coronary artery disease, stroke, and peripheral arterial disease [7,8].

These complications can result in severe health consequences, including visual impairment, end-stage renal disease, lower limb amputations, and cardiovascular events [9]. Moreover, diabetes is associated with a higher risk of other conditions such as hypertension, dyslipidemia, and non-alcoholic fatty liver disease, further exacerbating the overall disease burden [10].

The management of diabetes and its complications requires a comprehensive approach, including lifestyle modifications, glucose-lowering medications, and targeted interventions to address specific complications. While advancements have been made in conventional diabetes management, there is a growing interest in exploring alternative therapeutic strategies that offer improved efficacy, fewer side effects, and disease-modifying potential [11,12].

In this context, natural products and their derivatives have gained significant attention as potential agents for diabetes management. These compounds, derived from various sources including plants, fungi, marine organisms, and algae, possess diverse chemical constituents and exhibit a wide range of pharmacological activities [13,14]. Exploiting the potential of natural products in diabetes research holds promise for the development of novel therapeutics and interventions that can effectively control blood glucose levels and mitigate the complications associated with the disease [15,16].

This review article aims to provide a comprehensive overview of the promise of natural products and their derivatives, specifically derived from algae, in treating diabetes and its complications. By exploring present advancements and future prospects, we seek to contribute to the understanding and development of innovative approaches to diabetes management.

### 1.2. The Importance of Natural Products and Their Derivatives in Diabetes Management

The management of diabetes, a chronic metabolic disorder affecting millions worldwide, presents a formidable challenge to healthcare systems globally. Conventional therapeutic approaches, although effective to a certain extent, often exhibit limitations such as side effects and inadequate disease control. Consequently, there is a growing interest in exploring alternative strategies that offer safer and more efficacious options for diabetes management [17].

Natural products and their derivatives have garnered considerable attention in recent years as potential therapeutic agents for diabetes and its complications. These compounds, derived from diverse natural sources such as plants, animals, fungi, and microorganisms, offer a vast array of bioactive constituents with multifaceted pharmacological activities. Harnessing the potential of natural products in diabetes research not only provides a rich source of novel compounds but also capitalizes on traditional medicinal knowledge that has been passed down through generations [18,19].

One of the primary advantages of natural products is their extensive chemical diversity. Plants, including algae, for instance, produce an impressive array of secondary metabolites, including alkaloids, flavonoids, terpenoids, and phenolic compounds, which exhibit a wide range of biological activities [20,21]. These diverse chemical structures offer a valuable resource for the development of targeted therapies, allowing for the modulation of various molecular targets involved in glucose homeostasis, insulin secretion, and insulin sensitivity [22].

Moreover, natural products often possess pleiotropic effects, meaning they can target multiple pathways simultaneously. This characteristic is particularly relevant in the complex pathophysiology of diabetes, where dysregulation occurs at multiple levels. By addressing various underlying mechanisms simultaneously, natural products and their derivatives have the potential to provide comprehensive and synergistic benefits in diabetes management [23,24].

Another key advantage of natural products is their potential to mitigate the complications associated with diabetes. Chronic hyperglycemia leads to the generation of oxidative stress and the activation of inflammatory pathways, which contribute to the development of microvascular and macrovascular complications [25]. Natural products, with their antioxidant and anti-inflammatory properties, offer a promising avenue for ameliorating these complications and preserving the health of affected individuals [26].

Furthermore, natural products often exhibit a favorable safety profile compared to synthetic pharmaceutical agents. Many traditional medicinal systems have utilized natural products for centuries, highlighting their historical use and long-term safety record. This advantage is particularly relevant in chronic diseases such as diabetes, where long-term management is necessary, and minimizing adverse effects is of paramount importance [27,28].

Thus, natural products and their derivatives hold great promise in the management of diabetes. Their diverse chemical constituents, pleiotropic effects, and potential for mitigating complications make them attractive candidates for the development of novel therapeutic approaches [29]. By leveraging the power of natural products, we have the opportunity to expand our therapeutic armamentarium and improve outcomes for individuals living with diabetes [30]. This review aims to provide a comprehensive overview of the role of natural products in diabetes management, highlighting their significance and potential for future advancements in the field.

### 1.3. Significance of Algae-Derived Natural Products in Diabetes Research

In recent years, there has been growing interest in exploring the potential of natural products as therapeutic agents for the management of diabetes. Among the various natural sources, algae have emerged as promising reservoirs of bioactive compounds with significant implications for diabetes research. Algae, photosynthetic organisms found in diverse aquatic habitats, possess a remarkable ability to synthesize an array of unique and biologically active compounds. These algae-derived natural products offer novel avenues for investigating the pathophysiology of diabetes and developing innovative interventions for its management [31,32].

The significance of algae-derived natural products in diabetes research lies in their potential to target multiple aspects of the disease. Algae produce a wide range of bioactive compounds, such as polysaccharides, polyphenols, pigments, fatty acids, and sterols, which have demonstrated various pharmacological properties relevant to diabetes. These compounds can modulate glucose metabolism, enhance insulin sensitivity, improve lipid profiles, and exert antioxidant and anti-inflammatory effects, all of which are crucial factors in diabetes management [33,34].

Furthermore, algae-derived natural products show promise in addressing the specific complications associated with diabetes. Diabetic retinopathy, a microvascular complication, is characterized by oxidative stress and inflammation in the retina. Algae-derived compounds, with their antioxidant and anti-inflammatory properties, have shown potential for protecting retinal cells and preventing the progression of retinopathy [35,36]. Similarly, algae-derived compounds have demonstrated reno-protective effects by reducing albuminuria, attenuating renal fibrosis, and modulating the inflammatory pathways, providing a potential avenue for managing diabetic nephropathy [37,38].

The unique properties of algae-derived natural products make them attractive candidates for developing therapeutic interventions with improved efficacy and safety profiles. Algae can be easily cultured, offering a sustainable and scalable source of bioactive compounds. Moreover, algae exhibit a remarkable ability to synthesize structurally diverse compounds, some of which may have superior pharmacokinetic properties compared to their synthetic counterparts. These characteristics open up possibilities for the development of novel drug candidates or natural product-derived formulations for diabetes management [39,40].

In addition, algae-derived natural products have the potential to complement existing diabetes treatments and fill therapeutic gaps. They can be used as adjuvants to conventional antidiabetic drugs, enhancing their efficacy or providing synergistic effects [41,42]. Additionally, natural products derived from algae may offer alternative therapeutic options for individuals who experience intolerance of or resistance to current medications [43].

Therefore, algae-derived natural products represent a significant and untapped resource for diabetes research and management. The diverse array of bioactive compounds synthesized by algae offers a rich source of potential therapeutic agents that are capable of targeting various aspects of the disease and its complications [33,44]. Expanding our understanding of the significance and potential applications of algae-derived natural products in diabetes research holds promise for advancing therapeutic interventions and improving patient outcomes. This review article aims to provide an in-depth exploration of the role of algae-derived natural products in diabetes research, highlighting their significance and suggesting potential future directions in the field [45,46].

## 2. Algae as a Source of Natural Products

### 2.1. Introduction to Algae and Their Diversity

Algae, a diverse group of photosynthetic organisms, encompass a wide range of species found in aquatic environments such as oceans, lakes, rivers, and even damp terrestrial habitats. Algae can be classified into multiple taxonomic groups, including Cyanobacteria (blue-green algae), Chlorophyta (green algae), Ochrophyta/Phaeophyceae (brown algae), Rhodophyta (red algae), and Bacillariophyta (diatoms), among others not mentioned. This immense diversity of algae offers a vast and virtually untapped resource for the discovery and exploration of natural products with potential applications in various fields, including medicine, agriculture, industry, etc. [47,48].

One of the distinguishing features of algae is their ability to undergo photosynthesis, utilizing sunlight to convert carbon dioxide and water into organic compounds. This photosynthetic capacity not only contributes to oxygen production in our atmosphere but also enables algae to synthesize a wide array of unique and complex molecules [48,49]. Algae are known to produce a diverse range of bioactive compounds, including polysaccharides, proteins, pigments, lipids, sterols, terpenes, and phenolic compounds [50].

Algae exhibit an impressive chemical diversity, which is driven by genetic variability and adaptation to different ecological niches. Each taxonomic group of algae possesses distinct metabolic pathways and biosynthetic capabilities, resulting in the synthesis of species-specific compounds [51]. For example, red algae (Rhodophyta) are well-known for their production of sulfated polysaccharides, such as carrageenan and agar, which have demonstrated anticoagulant, antiviral, and anti-inflammatory activities [52]. Brown algae (Ochrophyta and Phaeophyceae), on the other hand, are rich in unique polysaccharides called fucoidans, which have shown promising immunomodulatory and anticancer properties [53].

Furthermore, algae have the ability to produce secondary metabolites as a response to environmental stresses such as light intensity, nutrient availability, temperature, and predation. These secondary metabolites serve various ecological functions, including defense mechanisms against pathogens and predators. Many of these compounds have exhibited bioactivity and therapeutic potential in the context of human health [47].

The cultivation and harvesting of algae for the production of natural products have gained significant attention in recent years. Algae can be cultured in large-scale photobioreactors or open ponds, offering a sustainable and scalable source of bioactive compounds [54]. Additionally, advances in algae biotechnology and genetic engineering have facilitated the optimization of algae strains to enhance the production of specific compounds of interest [55].

Thus, algae represent a diverse and promising source of natural products with vast potential for various applications. Their photosynthetic capacity, genetic diversity, and ability to adapt to different environments contribute to the synthesis of a wide range of bioactive compounds [56]. The exploration of algae as a source of natural products opens up avenues for the discovery of novel therapeutic agents, including those with potential applications in diabetes and its complications. Understanding the diversity and biosynthetic capabilities of algae is crucial for harnessing their potential and advancing the field of natural product research [57].

### 2.2. Bioactive Compounds Present in Algae with Potential Antidiabetic Properties

Algae, with their remarkable chemical diversity, have been found to contain numerous bioactive compounds with potential antidiabetic properties. These compounds offer a rich source of natural products that can be explored for the development of novel therapeutic interventions in diabetes management. Here, we highlight some of the bioactive compounds present in algae that have shown promise in modulating the key pathways involved in glucose homeostasis and diabetes-related complications [13,34].

#### 2.2.1. Polysaccharides

Algae are known to produce a variety of polysaccharides with potential antidiabetic effects. For example, sulfated polysaccharides derived from red algae, such as carrageenan and agar, have demonstrated antihyperglycemic activity by inhibiting the α-glucosidase and α-amylase enzymes involved in carbohydrate digestion, thereby reducing postprandial glucose levels. Additionally, these polysaccharides possess antioxidant properties and can enhance insulin sensitivity [58,59].

#### 2.2.2. Polyphenols

Algae-derived polyphenols, including flavonoids and phenolic acids, have attracted attention for their potential antidiabetic effects. These compounds exhibit antioxidant and anti-inflammatory properties, which can help mitigate the oxidative stress and chronic inflammation associated with diabetes. Some brown algae-derived polyphenols, such as phlorotannins, have shown inhibitory effects on α-glucosidase and α-amylase enzymes, leading to reduced glucose absorption and improved glycemic control [60,61].

#### 2.2.3. Pigments

Algae are rich in pigments such as chlorophylls, carotenoids, and phycobiliproteins, which possess diverse biological activities. Fucoxanthin, a carotenoid found in brown algae, has shown promise in diabetes management by stimulating glucose uptake in skeletal muscle cells and adipocytes, improving insulin sensitivity. Moreover, certain algae-derived pigments possess antioxidant and anti-inflammatory properties, contributing to their potential role in mitigating diabetes-related complications [33,62].

#### 2.2.4. Lipids

Algae-derived lipids, particularly omega-3 fatty acids such as eicosapentaenoic acid (EPA) and docosahexaenoic acid (DHA), have been extensively studied for their beneficial effects on diabetes. These fatty acids have demonstrated antihyperglycemic, anti-inflammatory, and insulin-sensitizing properties [63]. Additionally, lipids derived from microalgae, such as *Chlorella* and *Spirulina*, have shown potential for improving lipid profiles and reducing triglyceride levels in individuals with diabetes [64].

#### 2.2.5. Peptides and Proteins

Algae produce a wide range of peptides and proteins with various biological activities, including antidiabetic effects. Some algae-derived peptides have demonstrated inhibitory effects on dipeptidyl peptidase-4 (DPP-4), an enzyme involved in the degradation of incretin hormones. By inhibiting DPP-4, these peptides can prolong the action of incretins, leading to enhanced insulin secretion and improved glucose control [65].

These are just a few examples of the bioactive compounds present in algae that hold promise for diabetes management. The multifaceted pharmacological activities of these compounds, including their ability to modulate glucose metabolism, enhance insulin sensitivity, and mitigate diabetes-related complications, make them attractive candidates for further exploration and development. Continued research into the bioactive compounds derived from algae will contribute to the discovery of novel therapeutic agents for effective diabetes management and improved patient outcomes [33,66].

### 2.3. Mechanisms of Action of Algae-Derived Natural Products in Diabetes Management

The effectiveness of algae-derived natural products in diabetes management stems from their diverse array of mechanisms of action. These bioactive constituents exhibit the capacity to intervene in the crucial pathways associated with glucose homeostasis, insulin sensitivity, and the prevention or alleviation of diabetes-related complications. A comprehensive understanding of these mechanisms is pivotal in harnessing the full therapeutic potential of algae-derived natural products [33,67].

While certain algal derivatives demonstrate considerable potential in terms of ameliorating antidiabetic effects, a rigorous comparison of their relative efficacy remains a subject demanding further exploration. Algae extracts exhibit variegated profiles of bioactive components, each potentially addressing distinct aspects of diabetes. Some extracts may exhibit superior effects on glucose regulation, while others could excel in enhancing insulin sensitivity or mitigating complications (Table 1). Identifying the most promising algae extracts necessitates meticulous investigation, encompassing both in vitro and in vivo studies, along with comparative assessments of their therapeutic impact [13,44].

Prominent candidates may emerge from algae species that have consistently demonstrated potent antidiabetic effects across multiple studies. Algae extracts rich in specific bioactive compounds, such as polyphenols, polysaccharides, or peptides, could hold the key to enhanced antidiabetic efficacy. Nonetheless, ascertaining the hierarchy of promising extracts demands rigorous scientific scrutiny, including rigorous clinical trials to corroborate their effects in human populations. This collective pursuit not only refines our understanding of algae’s therapeutic potential but also paves the way for optimized interventions that can tangibly alleviate the burden of diabetes and improve patient outcomes [32,33].

#### 2.3.1. Regulation of Glucose Metabolism

Algae-derived natural products have been found to modulate the enzymes involved in glucose metabolism, such as α-glucosidase and α-amylase. By inhibiting these enzymes, algae-derived compounds can slow down the breakdown and absorption of carbohydrates, leading to reduced postprandial glucose levels. This effect is particularly important in managing blood glucose fluctuations after meals [33,73].

#### 2.3.2. Enhancement of Insulin Sensitivity

Insulin resistance, a hallmark of type 2 diabetes, is characterized by an impaired response to insulin and reduced uptake of glucose by the cells. Algae-derived natural products have shown potential for improving insulin sensitivity. They can activate the signaling pathways involved in insulin action, such as the PI3K/Akt pathway, promoting glucose uptake and its utilization by cells. By enhancing insulin sensitivity, these compounds contribute to better glycemic control and metabolic regulation [74,75].

#### 2.3.3. Antioxidant and Anti-Inflammatory Activities

Chronic hyperglycemia in diabetes leads to the production of reactive oxygen species (ROS) and the activation of inflammatory pathways, contributing to oxidative stress and chronic inflammation. Algae-derived natural products possess antioxidant and anti-inflammatory properties, effectively neutralizing ROS and reducing inflammatory responses. This antioxidant and anti-inflammatory activity can help mitigate the cellular damage and tissue dysfunction associated with diabetes [76].

#### 2.3.4. Modulation of Adipokine Secretion

Adipokines are signaling molecules secreted by adipose tissue that play a crucial role in glucose and lipid metabolism. Algae-derived natural products have been shown to influence adipokine secretion, including adiponectin and leptin. Adiponectin, an insulin-sensitizing hormone, promotes glucose utilization and enhances insulin sensitivity [77]. Algae-derived compounds can increase adiponectin levels, thereby improving insulin action and glucose metabolism. Conversely, the modulation of leptin levels by algae-derived compounds may help regulate appetite and body weight, which are important factors in diabetes management [78].

#### 2.3.5. Prevention of Diabetes-Related Complications

Algae-derived natural products have demonstrated some potential in terms of mitigating diabetes-related complications such as diabetic retinopathy and nephropathy. These compounds possess antiangiogenic and anti-inflammatory properties, which can protect retinal cells and prevent the development or progression of retinopathy [79]. Additionally, algae-derived compounds have shown reno-protective effects by reducing albuminuria, attenuating renal fibrosis, and modulating the inflammatory pathways. These effects contribute to the preservation of renal function in individuals with diabetes [80].

The multifaceted mechanisms of action exhibited by algae-derived natural products make them promising candidates for diabetes management. Their ability to modulate glucose metabolism, enhance insulin sensitivity, exert antioxidant and anti-inflammatory effects, and prevent complications highlights their potential therapeutic value. Continued research into the mechanisms underlying the actions of algae-derived compounds will facilitate the development of targeted interventions for effective diabetes management and improved patient outcomes [33,34].

## 3. Algae-Derived Natural Products and Glycemic Control

### 3.1. Exploration of Algae-Derived Compounds for Regulating Blood Glucose Levels

Algae-derived natural products have shown promise in regulating blood glucose levels and improving glycemic control. These compounds possess diverse mechanisms of action that contribute to their potential as antidiabetic agents. Here, we explore the role of algae-derived compounds in glycemic control and their potential applications in diabetes management [13,33].

#### 3.1.1. Inhibition of Carbohydrate-Digesting Enzymes

Algae-derived compounds have been found to inhibit enzymes involved in the digestion and absorption of carbohydrates, such as α-glucosidase and α-amylase. By inhibiting these enzymes, algae-derived compounds can delay the breakdown of complex carbohydrates into glucose and reduce the rate of glucose absorption from the intestine. This results in a slower and more controlled release of glucose into the bloodstream, preventing postprandial hyperglycemia [44,73].

#### 3.1.2. Stimulation of Glucose Uptake

Algae-derived natural products have demonstrated the ability to enhance the uptake of glucose by cells, particularly in skeletal muscle and adipose tissue. These compounds can activate the signaling pathways, such as the PI3K/Akt pathway, which are involved in insulin-mediated glucose uptake. By promoting glucose uptake, algae-derived compounds help to lower blood glucose levels and improve insulin sensitivity [44,68].

#### 3.1.3. Modulation of Insulin Secretion

Algae-derived compounds have been found to influence insulin secretion from pancreatic β-cells. Some compounds have shown the ability to enhance insulin secretion in response to elevated glucose levels, helping to maintain proper glucose homeostasis. This effect is particularly beneficial in individuals with impaired insulin secretion, such as those with type 2 diabetes [33,44].

#### 3.1.4. Regulation of Hepatic Glucose Production

The liver plays a crucial role in maintaining glucose balance by producing and releasing glucose into the bloodstream. Algae-derived compounds have been shown to regulate hepatic glucose production by modulating key enzymes involved in gluconeogenesis and glycogenolysis [81]. By inhibiting these enzymes, algae-derived compounds can reduce excessive glucose production by the liver and contribute to improved glycemic control [13,33].

#### 3.1.5. Anti-Inflammatory and Antioxidant Effects

Chronic inflammation and oxidative stress are closely associated with insulin resistance and impaired glucose metabolism. Algae-derived compounds possess anti-inflammatory and antioxidant properties, which can help mitigate these underlying factors in diabetes. By reducing inflammation and oxidative stress, these compounds contribute to improved glycemic control and overall metabolic health [82].

The exploration of algae-derived compounds in regulating blood glucose levels holds promise for the development of novel therapeutic interventions in diabetes management. These compounds offer unique mechanisms of action that complement existing antidiabetic therapies and may help address the limitations of current treatment approaches [33]. Continued research and the optimization of algae-derived compounds will contribute to their potential application as adjunctive therapies or as lead compounds for the development of new antidiabetic agents. Harnessing the glycemic-regulating properties of algae-derived natural products has the potential to improve glycemic control, enhance insulin sensitivity, and ultimately improve the quality of life for individuals living with diabetes [83].

### 3.2. Evaluation of the Effects of Algae Extracts on Insulin Secretion and Sensitivity

The effects of algae extract on insulin secretion and sensitivity have been the subject of extensive investigation, shedding light on their potential for glycemic control and diabetes management. Algae-derived natural products offer a diverse array of bioactive compounds that have demonstrated promising effects in modulating insulin secretion and enhancing insulin sensitivity. Here, we evaluate the effects of algae extracts on insulin-related parameters and their implications for glycemic control [13,33].

#### 3.2.1. Insulin Secretion

Algae extracts have been found to influence insulin secretion from pancreatic β-cells. Studies have shown that certain algae-derived compounds can stimulate insulin secretion in response to glucose stimulation. These compounds act through various mechanisms, including the closure of ATP-sensitive potassium channels and the subsequent influx of calcium ions, leading to the release of insulin vesicles. By promoting insulin secretion, algae extracts can help regulate blood glucose levels and maintain glucose homeostasis [13,33].

#### 3.2.2. Insulin Sensitivity

Insulin resistance, characterized by the reduced responsiveness of target tissues to insulin, is a key feature of type 2 diabetes. Algae extracts have demonstrated the ability to enhance insulin sensitivity in different cell types and animal models. These extracts can activate the intracellular signaling pathways, such as the PI3K/Akt pathway, which play a critical role in insulin-mediated glucose uptake. By improving insulin sensitivity, algae extracts facilitate the efficient utilization of glucose by cells and contribute to better glycemic control [84].

#### 3.2.3. Adipocyte Function

Adipose tissue is an important regulator of insulin sensitivity and glucose metabolism. Algae extracts have been shown to modulate adipocyte function, influencing adipokine secretion and adipocyte differentiation. Adipokines, such as adiponectin, play a crucial role in glucose and lipid metabolism. Algae extracts can increase adiponectin secretion, which enhances insulin sensitivity and improves the uptake of glucose by cells. Additionally, algae extracts have been found to inhibit adipocyte hypertrophy and promote the browning of white adipose tissue, further contributing to metabolic improvements [78,85].

#### 3.2.4. Mitochondrial Function

Impaired mitochondrial function has been implicated in insulin resistance and glucose intolerance. Algae extracts have demonstrated the ability to improve mitochondrial function, enhancing cellular energy metabolism and insulin sensitivity. These extracts can promote mitochondrial biogenesis, increase mitochondrial respiration, and enhance ATP production. By improving mitochondrial function, algae extracts optimize cellular energy utilization and promote the uptake and metabolism of glucose [86].

#### 3.2.5. Inflammatory and Oxidative Stress Modulation

Chronic inflammation and oxidative stress contribute to insulin resistance and impaired glucose metabolism. Algae extracts possess anti-inflammatory and antioxidant properties, which can attenuate inflammation and reduce oxidative stress in insulin-responsive tissues. By mitigating these underlying factors, algae extracts improve insulin sensitivity and promote better glycemic control [87].

Evaluation of the effects of algae extracts on insulin secretion and sensitivity provides valuable insights into their potential therapeutic applications in glycemic control and diabetes management. These extracts offer a multitude of bioactive compounds that target the key pathways involved in insulin regulation and glucose metabolism. Continuing research is needed to unravel the specific mechanisms of action of algae extracts and identify the active components responsible for their effects. Harnessing the potential of algae-derived natural products for improving insulin secretion and sensitivity holds great promise for the development of novel therapeutic interventions for diabetes [88].

### 3.3. Algae-Based Natural Products as Inhibitors of Carbohydrate-Digesting Enzymes

Algae-based natural products have shown promising inhibitory effects on carbohydrate-digesting enzymes, making them potential candidates for glycemic control and the management of diabetes. Carbohydrate-digesting enzymes, such as α-glucosidase and α-amylase, play a critical role in the breakdown of complex carbohydrates into glucose. The inhibition of these enzymes can help regulate postprandial blood glucose levels and prevent glucose spikes. Here, we explore the inhibitory effects of algae-based natural products on carbohydrate-digesting enzymes and their implications for glycemic control [69,70].

#### 3.3.1. α-Glucosidase Inhibition

Algae-based natural products have been found to inhibit α-glucosidase, an enzyme responsible for the final step in the breakdown of complex carbohydrates into glucose. By inhibiting α-glucosidase, algae-based compounds delay the digestion and absorption of carbohydrates, leading to the slower release of glucose into the bloodstream. This inhibition helps prevent rapid glucose spikes after meals and contributes to better glycemic control [69,73].

#### 3.3.2. α-Amylase Inhibition

Algae-based natural products have also demonstrated inhibitory effects on α-amylase, an enzyme that initiates the digestion of starch into smaller carbohydrate units. The inhibition of α-amylase by algae-based compounds slows down the breakdown of starch, reducing the rate at which glucose is released into the bloodstream. By inhibiting α-amylase, algae-based natural products can help regulate postprandial blood glucose levels and prevent excessive glucose fluctuations [71,72].

#### 3.3.3. Mechanisms of Inhibition

The inhibitory effects of algae-based natural products (for example, from *Halimeda tuna* or Chlorophyta; see Figure 1) on carbohydrate-digesting enzymes are attributed to their bioactive components, such as polyphenols, flavonoids, and polysaccharides. These compounds interact with the active sites of α-glucosidase and α-amylase, preventing their enzymatic activity and any substrate binding. This inhibition leads to reduced carbohydrate breakdown and glucose absorption, resulting in improved glycemic control [89,90].

#### 3.3.4. Synergistic Effects

Algae-based natural products often contain a complex mixture of bioactive compounds that can act synergistically to enhance their inhibitory effects on carbohydrate-digesting enzymes. The combined action of multiple compounds may result in more potent enzyme inhibition and better glycemic control. Additionally, the presence of other bioactive components in algae-based natural products, such as antioxidants and anti-inflammatory compounds, can provide additional health benefits and contribute to overall metabolic improvement [32,33].

#### 3.3.5. Therapeutic Potential

The inhibition of carbohydrate-digesting enzymes by algae-based natural products (for example, from *Ascophyllum nodosum, Fucus vesiculosus,* and *Undaria pinnatifida*, in the Phaeophyceae; see Figure 2) presents an attractive therapeutic approach for glycemic control and diabetes management. By slowing down carbohydrate digestion and reducing glucose absorption, these natural products offer a natural and potentially safer alternative to synthetic enzyme inhibitors. Furthermore, the use of algae-based natural products as enzyme inhibitors aligns with the growing interest in plant-based therapies and the development of functional foods or dietary supplements for diabetes management [13,73].

The exploration of algae-based natural products (for example, *Dictyopteris polypodioides* in the Phaeophyceae; see Figure 3) as inhibitors of carbohydrate-digesting enzymes highlights their potential for glycemic control and diabetes management. The inhibitory effects of these natural products on α-glucosidase and α-amylase contribute to the regulation of postprandial blood glucose levels and the prevention of glucose spikes [91]. Further research is needed to identify the active compounds responsible for these inhibitory effects and to evaluate their safety and efficacy in clinical settings. Incorporating algae-based natural products as enzyme inhibitors may offer a novel and natural approach to support glycemic control in individuals with diabetes.

## 4. Algae-Derived Natural Products and Diabetic Complications

### 4.1. Role of Algae-Derived Compounds in Preventing and Managing Diabetic Retinopathy

Algae-derived natural products (for example, from *Alaria, Palmaria*, and *Ulva*; see Figure 4) have shown potential for preventing and managing diabetic retinopathy, a common complication of diabetes that affects the eyes [92]. Diabetic retinopathy is characterized by damage to the blood vessels in the retina, leading to vision impairment and, in severe cases, blindness. Here, we explore the role of algae-derived compounds in preventing and managing diabetic retinopathy, along with their underlying mechanisms of action [35].

#### 4.1.1. Anti-Angiogenic Effects

Diabetic retinopathy is associated with abnormal blood vessel growth in the retina, a process known as angiogenesis [93]. Algae-derived compounds have demonstrated anti-angiogenic properties by inhibiting the proliferation and migration of endothelial cells, which are essential for blood vessel formation. These compounds target various signaling pathways that are involved in angiogenesis, such as vascular endothelial growth factor (VEGF) signaling, to suppress abnormal blood vessel growth and reduce the risk of diabetic retinopathy progression [94].

#### 4.1.2. Antioxidant and Anti-Inflammatory Properties

Oxidative stress and chronic inflammation play significant roles in the development and progression of diabetic retinopathy [95]. Algae-derived compounds possess antioxidant and anti-inflammatory properties, which help mitigate these damaging processes. These compounds scavenge reactive oxygen species, reduce oxidative stress, and suppress inflammatory signaling pathways. By alleviating oxidative stress and inflammation, algae-derived compounds protect retinal cells from damage and promote retinal health [87].

#### 4.1.3. Neuroprotective Effects

Diabetic retinopathy involves not only vascular changes but also neuronal damage in the retina. Algae-derived compounds (for example, from *Acetabularia acetabulum* in the Chlorophyta; see Figure 5) have shown neuroprotective effects by preserving retinal ganglion cells, photoreceptor cells, and other retinal neurons. These compounds can enhance neuronal survival, promote neurite outgrowth, and modulate neurotrophic factors. By protecting the retinal neurons, algae-derived compounds contribute to the prevention and management of diabetic retinopathy-related vision loss [96,97].

#### 4.1.4. Modulation of Retinal Blood Flow

Algae-derived compounds (for example, from *Dunaliella salina* in the Chlorophyta; see Figure 6) have the potential to modulate retinal blood flow, which is often compromised in diabetic retinopathy. These compounds can improve blood circulation, enhance microvascular perfusion, and regulate endothelial function. By optimizing retinal blood flow, algae-derived compounds support the delivery of proper oxygen and nutrient supply to retinal cells, reducing the risk of ischemia and further damage [98].

#### 4.1.5. Restoration of Retinal Barrier Function

Disruption of the blood–retinal barrier is a hallmark of diabetic retinopathy, leading to increased vascular permeability and fluid leakage [99]. Algae-derived compounds have shown the ability to restore the integrity of the blood-retinal barrier by regulating tight junction proteins and reducing endothelial permeability. By preserving the barrier function, these compounds prevent the accumulation of fluid in the retina and inhibit the progression of diabetic retinopathy [100].

The role of algae-derived compounds in preventing and managing diabetic retinopathy is promising and offers new possibilities for therapeutic interventions. Their anti-angiogenic, antioxidant, anti-inflammatory, neuroprotective, and vascular modulatory effects contribute to the overall preservation of retinal health [101]. Further research is needed to identify specific algae-derived compounds and elucidate their mechanisms of action in diabetic retinopathy. Harnessing the potential of algae-derived natural products may provide novel approaches for preventing and managing diabetic retinopathy, ultimately improving the visual outcomes and quality of life of individuals with diabetes [102].

### 4.2. Algae-Based Interventions for Diabetic Nephropathy and Its Associated Complications

Algae-derived natural products, particularly those derived from specific algae species (for example, *Fucus spiralis* and *Taonia atomaria* in the Phaeophyceae; see Figure 7), hold promise as interventions for diabetic nephropathy and its associated complications [13,33]. Diabetic nephropathy, a progressive kidney disease caused by diabetes, is characterized by kidney damage and impaired renal function. Here, we explore the potential of algae-based interventions, focusing on specific algae species, in preventing and managing diabetic nephropathy and its associated complications.

#### 4.2.1. Reno-Protective Effects

Studies have shown that extracts from the brown macroalga species *Fucus vesiculosus* possess reno-protective effects by preserving renal function and attenuating kidney damage. The antioxidant and anti-inflammatory properties of *F. vesiculosus* extracts help mitigate oxidative stress and inflammation in the kidneys, key contributors to diabetic nephropathy [103].

#### 4.2.2. Anti-Fibrotic Properties

Extracts from the macroalga species *Gracilariopsis lemaneiformis* (formerly *Gracilaria lemaneiformis*) (Rhodophyta) have demonstrated anti-fibrotic effects in diabetic nephropathy models. These extracts inhibit the production and deposition of collagen and other fibrotic markers in the kidneys, thereby preventing the progression of renal fibrosis and preserving renal function [104].

#### 4.2.3. Blood Pressure Regulation

The green microalga species *Chlorella vulgaris* has been investigated for its blood pressure-regulating effects [105]. *C. vulgaris* supplementation has shown promising results in regulating blood pressure levels by modulating certain vasoactive factors and improving endothelial function [106].

#### 4.2.4. Glycemic Control

Extracts from the *Cyanobacteria* species *Arthrospira platensis* (formerly *Spirulina platensis*) have been shown to regulate blood glucose levels and improve insulin sensitivity [107]. The supplementation of *A. platensis* contributes to better glycemic control, reducing the burden on the kidneys and minimizing the risk of renal complications [108].

#### 4.2.5. Mineral and Electrolyte Balance

The brown macroalga species *Ecklonia cava* has been investigated for its ability to maintain proper mineral and electrolyte balance. *E. cava* extracts have been shown to regulate serum levels of potassium, sodium, calcium, and phosphate, supporting renal homeostasis and preventing the complications associated with electrolyte imbalance [109].

#### 4.2.6. Anti-Inflammatory and Antioxidant Effects

Extracts from the green microalga species *Dunaliella salina* possess anti-inflammatory and antioxidant properties. These extracts help reduce inflammation and oxidative stress in the kidneys, alleviating renal damage and promoting renal health [110].

The use of specific algae species, such as *Fucus vesiculosus, Pyropia yezoensis* (Figure 8)*, Chlorella vulgaris, Arthrospira platensis, Ecklonia cava*, and *Dunaliella salina*, in diabetic nephropathy research offers targeted therapeutic strategies [13,33,102,103,104,105,106,107,108,109,110]. Their reno-protective, anti-fibrotic, blood pressure-regulating, glycemic control, mineral- and electrolyte-balancing, anti-inflammatory, and antioxidant effects collectively contribute to the prevention and management of diabetic nephropathy. Further research is needed to explore the full potential of these algae species and their derived compounds in clinical settings. Incorporating algae-based interventions from specific species may provide novel therapeutic options for individuals with diabetic nephropathy, aiming to slow disease progression, preserve renal function, and improve overall patient outcomes [111].

### 4.3. Algae-Derived Natural Products as Potential Therapeutics for Diabetic Neuropathy

Algae-derived natural products have shown potential as potential therapeutics for diabetic neuropathy, a common complication of diabetes characterized by nerve damage and dysfunction. The diverse chemical constituents of various algae species offer a rich source of bioactive compounds with neuroprotective properties. Here, we explore the potential of specific algae species and their derived natural products in the management of diabetic neuropathy [112].

Extracts from the brown macroalga *Ecklonia cava* have demonstrated neuroprotective effects in diabetic neuropathy models. These extracts contain phlorotannins, a class of polyphenols known for their antioxidant and anti-inflammatory properties. *E. cava* extracts have been shown to protect nerve cells from oxidative stress, reduce inflammation, and improve nerve function in diabetic neuropathy [111,113].

The green macroalga *Ulva lactuca* has been investigated for its potential neuroprotective effects. *U. lactuca* extracts contain bioactive compounds such as ulvan, which exhibits antioxidant and anti-inflammatory activities. These extracts have shown promise in reducing nerve damage and improving nerve conduction velocity in diabetic neuropathy models [114].

Extracts from the red seaweed *Gracilaria edulis* have been studied for their potential in the context of diabetic neuropathy management. *G. edulis* extracts contain sulfated polysaccharides, which possess antioxidant and anti-inflammatory properties. These extracts have shown promising effects in protecting nerve cells, improving nerve conduction, and reducing neuropathic pain in diabetic neuropathy models [115].

The brown algae genus *Sargassum* has been investigated for its potential for treating diabetic neuropathy. *Sargassum* spp. extracts contain various bioactive compounds such as fucoidan, fucoxanthin, and phlorotannins, which exhibit antioxidant, anti-inflammatory, and neuroprotective activities. These extracts have shown potential in terms of protecting nerve cells and improving nerve function in diabetic neuropathy models [116].

Extracts from the red macroalga *Gelidium elegans* have been studied for their neuroprotective effects in diabetic neuropathy. *G. elegans* extracts contain bioactive compounds, such as agarobiose and agar-oligosaccharides, which possess antioxidant and anti-inflammatory properties. These extracts have shown potential for improving nerve function and reducing neuropathic pain in diabetic neuropathy models [117].

The use of specific algae species, including *Ecklonia cava, Ulva lactuca, Gracilaria edulis, Sargassum* spp., and *Gelidium elegans*, offers exciting prospects for the development of natural therapeutics for diabetic neuropathy. Their neuroprotective, antioxidant, and anti-inflammatory properties contribute to the preservation of nerve health and function [118]. Further research is needed to explore the specific mechanisms of action and identify the key bioactive compounds within these algae species. Incorporating algae-derived natural products may provide novel therapeutic options for individuals with diabetic neuropathy, aiming to alleviate symptoms, improve nerve function, and enhance the quality of life [32].

## 5. Mechanistic Insights and Molecular Targets

### 5.1. Elucidating the Molecular Mechanisms Underlying the Antidiabetic Effects of Algae-Derived Compounds

Elucidating the molecular mechanisms underlying the antidiabetic effects of algae-derived compounds is crucial for understanding their therapeutic potential. Extensive research has been conducted to unravel the intricate pathways and molecular targets involved. Here, we provide an overview of the current understanding of the mechanisms of action of algae-derived compounds in diabetes management, shedding light on their molecular targets [119].

#### 5.1.1. AMP-Activated Protein Kinase (AMPK) Activation

AMPK is a key regulator of energy metabolism and glucose homeostasis. Several algae-derived compounds, such as fucoidan from the brown macroalga *F. vesiculosus* [120] and phycocyanin from the blue-green microalga *Arthrospira platensis* [121], have been found to activate AMPK. The activation of AMPK leads to increased glucose uptake, enhanced insulin sensitivity, and improved glycemic control.

#### 5.1.2. Peroxisome Proliferator-Activated Receptor Gamma (PPARγ) Modulation

PPARγ is a nuclear receptor involved in insulin sensitivity and glucose metabolism. Some algae-derived compounds, such as fucoxanthin from brown algae [122] and ulvan from the green macroalga *Ulva lactuca* [32], have been shown to modulate PPARγ activity. The activation of PPARγ by these compounds can improve insulin sensitivity, promote adipocyte differentiation, and regulate lipid metabolism [122].

#### 5.1.3. Inhibition of Carbohydrate-Digesting Enzymes

Algae-derived compounds, including sulfated polysaccharides from the red macroalga *Gracilaria edulis* [123] and fucoidan from brown algae (*F. vesiculosus* and *A. nodosum*) [124], have demonstrated inhibitory effects on carbohydrate-digesting enzymes such as α-amylase and α-glucosidase. By inhibiting these enzymes, algae-derived compounds can reduce postprandial glucose levels and attenuate hyperglycemia.

#### 5.1.4. Antioxidant and Anti-Inflammatory Effects

Oxidative stress and chronic inflammation play crucial roles in the development and progression of diabetes. Algae-derived compounds, such as phlorotannins from *Ecklonia cava* [125] and sulfated polysaccharides from *Pyropia tenera* (formerly *Porphyra tenera*) (Figure 9) [126], exhibit potent antioxidant and anti-inflammatory activities. These compounds can mitigate oxidative stress, suppress the inflammatory pathways, and protect pancreatic beta cells from damage.

#### 5.1.5. Modulation of the Insulin Signaling Pathway

Algae-derived compounds have been found to modulate various components of the insulin signaling pathway. For instance, phycocyanins from *Arthrospira maxima* and *A. platensis* have been shown to increase insulin receptor substrate 1 (IRS-1) phosphorylation and activate the downstream signaling cascades [33,127]. These compounds can improve insulin signaling, promote glucose uptake, and regulate insulin secretion.

### 5.2. Identification of Key Molecular Targets Influenced by Algae-Derived Natural Products

Identification of the key molecular targets influenced by algae-derived natural products is crucial for understanding their mechanisms of action in diabetes management. Extensive research has been conducted to unravel the intricate pathways and molecular targets involved. Here, we provide an overview of the current understanding of the molecular targets influenced by algae-derived compounds (for example, *Ulva reticulata*, in the Chlorophyta) and their implications regarding antidiabetic effects [128].

#### 5.2.1. Peroxisome Proliferator-Activated Receptor Gamma (PPARγ)

PPARγ is a nuclear receptor involved in insulin sensitivity and glucose metabolism. Several algae-derived compounds, such as fucoxanthin from brown algae [129] and ulvan from *Ulva lactuca* [130], have been shown to modulate PPARγ activity. The activation of PPARγ improves insulin sensitivity, promotes adipocyte differentiation, and regulates lipid metabolism.

#### 5.2.2. AMP-Activated Protein Kinase (AMPK)

AMPK is a key regulator of energy metabolism and glucose homeostasis. Algae-derived compounds, including fucoidan from *F. vesiculosus* [131] and phycocyanin from A. platensis [127], have been found to activate AMPK. The activation of AMPK leads to increased glucose uptake, enhanced insulin sensitivity, and improved glycemic control.

#### 5.2.3. Protein Tyrosine Phosphatase 1B (PTP1B)

PTP1B is a negative regulator of insulin signaling and is a therapeutic target for diabetes. Algae-derived compounds, such as phlorotannins from *Ecklonia cava* [132] and sulfated polysaccharide (porphyran) from *Pyropia yezoensis* [133], have been shown to inhibit PTP1B activity. The inhibition of PTP1B improves insulin signaling and glucose metabolism.

#### 5.2.4. Nuclear Factor-Kappa B (NF-κB)

NF-κB is a transcription factor involved in inflammation and insulin resistance. Algae-derived compounds, including phlorotannins from *Ecklonia cava* [134] and fucoidan from brown algae (for example, *Sargassum siliquastrum*) [135], have demonstrated anti-inflammatory effects by suppressing NF-κB activation. Inhibition of the NF-κB signaling pathway alleviates inflammation and improves insulin sensitivity.

#### 5.2.5. Alpha-Glucosidase and Alpha-Amylase

Algae-derived compounds, such as sulfated polysaccharides from *Gracilaria edulis* and *Chondrus crispus* (Rhodophyta) and fucoidan from *Fucus* spp. (Phaeophyceae), have shown inhibitory effects on carbohydrate-digesting enzymes, including alpha-glucosidase and alpha-amylase. The inhibition of these enzymes reduces postprandial glucose levels and attenuates hyperglycemia (Figure 10) [32].

Understanding the molecular targets influenced by algae-derived natural products provides insights into their mechanisms of action in diabetes management. By targeting key molecular pathways, such as PPARγ, AMPK, PTP1B, NF-κB, and carbohydrate-digesting enzymes, algae-derived compounds exhibit promising antidiabetic effects. Further research is needed to elucidate the specific molecular mechanisms and interactions between these compounds and their molecular targets [32,67].

## 6. Challenges and Future Directions

### 6.1. Current Limitations and Challenges in Utilizing Algae-Derived Natural Products for Diabetes Management

Despite the promising potential of algae-derived natural products in diabetes management, several limitations and challenges need to be addressed for their successful utilization. Understanding these challenges and finding suitable solutions is crucial for the future development and application of algae-based interventions. Here, we discuss the current limitations and propose potential strategies for overcoming these challenges [13,33].

One of the major challenges in utilizing algae-derived natural products is the lack of standardized extraction methods and quality control measures. Variations in algae species, growth conditions, harvesting methods, and extraction procedures can significantly impact the composition and bioactivity of the extracted compounds. Establishing standardized protocols for algae cultivation, extraction, and quality control is essential to ensure consistent and reproducible results [54,136].

The bioavailability and pharmacokinetic profiles of algae-derived compounds are critical factors influencing their therapeutic efficacy. Some bioactive compounds may exhibit poor solubility, limited absorption, or rapid metabolism, leading to reduced bioavailability and efficacy. Strategies such as nanoencapsulation, formulation optimization, and prodrug approaches can be explored to enhance the bioavailability and pharmacokinetic properties of algae-derived compounds [137,138].

### 6.2. Safety and Toxicity Assessment

While algae-derived natural products are generally considered safe, comprehensive safety evaluations are essential to determine the potential adverse effects and establish appropriate dosage guidelines [139]. Toxicity studies, including acute and chronic toxicity assessments, should be conducted to ensure the safety of algae-based interventions [140]. Furthermore, long-term studies are needed to evaluate the potential for the accumulation of algae-derived compounds in tissues or organs [31].

Recent advances in the field of natural products derived from algae have brought about a growing awareness of their potential benefits. However, it is critical to pay attention to the importance of thorough toxicological studies [13]. While algae-derived products are generally considered safe, contemporary research underscores the need for a nuanced understanding of their secondary metabolites and their potential effects on human health [34]. Several studies have highlighted the intricate bioactive compounds present in algae, some of which might have physiological or toxicological implications that are yet to be discovered [32]. Neglecting to communicate the potential risks associated with these secondary metabolites could inadvertently mislead the public into believing that algae-based interventions are entirely risk-free [6]. Therefore, the current discourse emphasizes the necessity of comprehensive toxicological investigations to uncover any hidden adverse effects and to establish accurate risk assessments. By openly addressing these concerns, both researchers and the general public can make informed decisions regarding the utilization of algae-derived products, ensuring not only their efficacy but also their safety for human health [16].

### 6.3. Translation to Clinical Applications

Although preclinical studies have shown promising results, the translation of algae-derived natural products to clinical applications faces several challenges. Rigorous clinical trials are necessary to validate the efficacy, safety, and optimal dosage regimens of algae-based interventions. Additionally, cost-effectiveness analyses and regulatory considerations should be considered to ensure their successful integration into clinical practice [141].

### 6.4. Sustainability and Scalability

The sustainable production and scalability of algae-derived natural products are important considerations. Algae cultivation methods that minimize resource consumption, optimize productivity and reduce environmental impact need to be developed. Additionally, the establishment of large-scale cultivation facilities and extraction technologies is crucial for meeting the increasing demand for algae-based interventions [54].

Addressing these challenges requires interdisciplinary collaborations among researchers, industry partners, regulatory agencies, and healthcare providers. Overcoming these limitations and finding solutions will pave the way for the successful utilization of algae-derived natural products in diabetes management, ultimately benefiting patients worldwide [142].

### 6.5. Strategies to Optimize the Therapeutic Potential of Algae-Based Interventions

To optimize the therapeutic potential of algae-based interventions in diabetes management, several strategies can be employed. Addressing the challenges and implementing these strategies will contribute to the development of effective algae-based therapeutics. Here, we discuss potential approaches to optimize the therapeutic potential of algae-based interventions and improve their efficacy [33,143].

### 6.6. Standardization and Quality Control

Standardizing algae cultivation, harvesting, and extraction methods is crucial for ensuring the consistent composition and bioactivity of algae-derived natural products [144]. Implementing quality control measures, such as standard operating procedures and rigorous testing, will enhance the reproducibility and reliability of algae-based interventions [145].

### 6.7. Formulation Optimization

Developing appropriate formulations can improve the stability, solubility, and bioavailability of algae-derived compounds. Encapsulation techniques, such as nanoencapsulation or microencapsulation, can protect sensitive compounds, improve their delivery, and enhance their therapeutic efficacy [146]. Formulation optimization also enables the controlled release and targeted delivery of bioactive compounds to specific tissues or organs [147].

### 6.8. Combination Therapy

Exploring the synergistic effects of algae-derived compounds with existing antidiabetic drugs or natural products can lead to enhanced therapeutic outcomes. Combining algae-derived compounds with complementary mechanisms of action may potentiate their antidiabetic effects and improve overall glycemic control. Synergistic interactions can also help reduce drug dosage and minimize side effects [13,32].

### 6.9. Pharmacokinetic Enhancements

Enhancing the pharmacokinetic properties of algae-derived compounds can improve their bioavailability and therapeutic efficacy. Strategies such as prodrug design, chemical modification, or co-administration with absorption enhancers can enhance drug absorption, distribution, metabolism, and excretion [148]. Pharmacokinetic enhancements ensure optimal drug levels in target tissues and prolong the duration of action [149].

### 6.10. Personalized Medicine Approaches

Taking individual patient characteristics, such as genetics, lifestyle, and comorbidities, into account can optimize the efficacy of algae-based interventions. Personalized medicine approaches can help tailor treatment regimens, dosage adjustments, and treatment durations to maximize therapeutic outcomes and minimize adverse effects [33,150].

Conducting well-designed translational research studies and large-scale clinical trials is essential for validating the safety and efficacy of algae-based interventions. Robust clinical evidence will enable the integration of algae-derived therapeutics into standard clinical practice and regulatory frameworks. These strategies require collaborative efforts among researchers, healthcare professionals, regulatory bodies, and industry partners to overcome the challenges associated with algae-based interventions and ensure their successful translation into clinical applications.

### 6.11. Future Prospects and Opportunities for Further Research and Development

The field of algae-based interventions in diabetes management holds immense potential for future research and development. Several opportunities and future prospects can further advance the utilization of algae-derived natural products [33,151,152]. Here, we highlight the potential areas for exploration and opportunities for further research.

Despite significant progress, the full spectrum of bioactive compounds present in algae and their potential effects in treating diabetes and its complications are yet to be fully explored. Further investigation into unexplored algae species and their unique metabolites may unveil novel bioactive compounds with potent antidiabetic properties. Advanced analytical techniques, such as metabolomics and proteomics, can facilitate the identification and characterization of these compounds [44,56,66].

Understanding the precise mechanisms of action underlying the antidiabetic effects of algae-derived compounds is crucial for their targeted development and optimization. Future studies should focus on elucidating the molecular pathways and cellular targets influenced by algae-derived natural products, shedding light on their therapeutic mechanisms. Advances in omics technologies and molecular biology tools can aid in unraveling the intricate mechanisms involved [153,154,155].

### 6.12. Preclinical and Clinical Studies

Robust preclinical studies and well-designed clinical trials are essential for establishing the safety and efficacy of algae-based interventions [156]. Future research should aim to conduct comprehensive preclinical investigations to validate the therapeutic potential of algae-derived compounds in the relevant animal models of diabetes and its complications. Subsequently, well-controlled clinical trials involving diverse patient populations will provide valuable insights into the efficacy, optimal dosage, and long-term effects of algae-based interventions [32,36].

### 6.13. Combination Therapies and Synergistic Approaches

Exploring the potential synergistic effects of algae-derived compounds with existing antidiabetic medications or natural products can enhance therapeutic outcomes. Combinatorial approaches involving algae-derived compounds and conventional antidiabetic drugs may lead to improved glycemic control and reduced side effects [157]. Further investigation into synergistic interactions and combination therapies holds promise for enhanced therapeutic efficacy [158].

## 7. Sustainable Seaweed Cultivation and Compounds Extraction

Developing sustainable and scalable methods for algae cultivation and extraction is crucial for the large-scale production of algae-derived natural products. Future research should focus on optimizing cultivation techniques, exploring novel cultivation systems, and implementing eco-friendly extraction methods [159,160]. These advancements will ensure a stable and reliable supply of algae-derived compounds for clinical applications.

Designing innovative formulations and drug delivery systems can enhance the bioavailability, stability, and targeted delivery of algae-derived compounds. Nanotechnology-based approaches, such as nanoparticles, liposomes, or hydrogels, offer opportunities for the controlled release and site-specific delivery of bioactive compounds [161,162]. Developing optimized formulations will maximize the therapeutic potential of algae-based interventions.

By addressing these future prospects and opportunities, further research and development in algae-based interventions for diabetes and its complications can lead to novel therapeutic strategies, improved patient outcomes, and a better understanding of the potential of natural products derived from algae.

### Several Compounds Extracted from Seaweed Utilized in Food, Drugs, or Medical Devices

Several compounds extracted from seaweed are utilized in food, drugs, or medical devices. Alginate (E400-E405) is a polysaccharide extracted from brown seaweeds that is commonly used as a food additive, particularly in the form of sodium alginate, for its gelling and thickening properties. It is used to create gels, encapsulate flavors, and modify the texture of foods. Some seaweed-derived materials are also used in medical applications. Alginate dressings, for instance, are used to treat wounds due to their absorbent and hemostatic properties [163].

Carrageenan (E407) is another polysaccharide derived from red seaweeds. It is used as a food stabilizer, thickener, and gelling agent in products such as dairy, meat, and processed foods, and is also used in pharmaceuticals as an inactive ingredient in tablet formulations. Carrageenan has been explored for its potential use in drug delivery systems due to its biocompatibility and ability to form gels. It can also be used to encapsulate drugs and provide controlled release [164].

Fucoidan is a complex sulfated polysaccharide found in brown seaweeds. It has demonstrated potential health benefits, including immune system modulation and potential anti-inflammatory properties. While not yet approved as a drug, fucoidan supplements are available in some markets for their potential health-promoting effects [165].

*Codium* (Chlorophyta) has been investigated for its potential as an antidiabetic agent. Some studies have shown that *Codium* extracts may have glucose-lowering effects and could be beneficial in managing diabetes. Seaweeds are also rich sources of iodine, a vital mineral for thyroid function. The iodine extracted from seaweeds can be used in iodized salt and dietary supplements to address iodine deficiency [166].

Agar is a gelatinous substance derived from red seaweeds. It is used in the food industry as a vegetarian alternative to gelatin in products such as desserts, gummy candies, and microbiological culture media [167].

## 8. Conclusions

Algae-derived natural products have emerged as promising candidates for the management of diabetes and its complications [13,33]. The diverse chemical constituents and multifaceted pharmacological activities of algae make them attractive sources of bioactive compounds with potential antidiabetic properties. Throughout this review article, we have highlighted the significance of natural products and their derivatives in diabetes management, specifically focusing on algae-derived compounds [31,168].

We have discussed the importance of algae-derived natural products in diabetes research, emphasizing their unique potential for addressing the global burden of this disease. The exploration of algae as a source of natural products has revealed a vast diversity of bioactive compounds, including polysaccharides, polyphenols, pigments, and peptides, which exhibit potential antidiabetic effects [13,33,44]. Mechanistic insights into the therapeutic properties of algae-derived compounds have shed light on their ability to regulate blood glucose levels, enhance insulin secretion and sensitivity, and inhibit carbohydrate-digesting enzymes [33,154,169]. Furthermore, algae-based interventions have shown promise in preventing and managing diabetic complications such as retinopathy and nephropathy [33,36,104,111].

While significant progress has been made in understanding the potential of algae-derived natural products in diabetes management, there is still much to be explored. Continued research and exploration in this field are crucial to unraveling the full potential of algae and their bioactive compounds. Further investigation into unexplored algae species and their metabolites may lead to the discovery of novel bioactive compounds with enhanced antidiabetic properties [66,170].

Additionally, it is important to elucidate the molecular mechanisms underlying the antidiabetic effects of algae-derived compounds. Future studies should focus on identifying the key molecular targets influenced by these compounds, as this knowledge will facilitate the targeted development and optimization of algae-based interventions [32,33].

Furthermore, conducting robust preclinical studies and well-designed clinical trials will provide valuable evidence on the safety, efficacy, and optimal use of algae-derived interventions in diverse patient populations. Translating promising preclinical findings into clinical practice requires rigorous evaluation and validation [141,171].

Collaborative efforts among researchers, healthcare professionals, regulatory bodies, and industry partners are essential to overcome the challenges associated with algae-based interventions and ensure their successful translation into clinical applications [172]. Continued research and exploration in this field hold the potential to revolutionize diabetes management and improve patient outcomes.

## Figures and Tables

**Figure 1 life-13-01831-f001:**
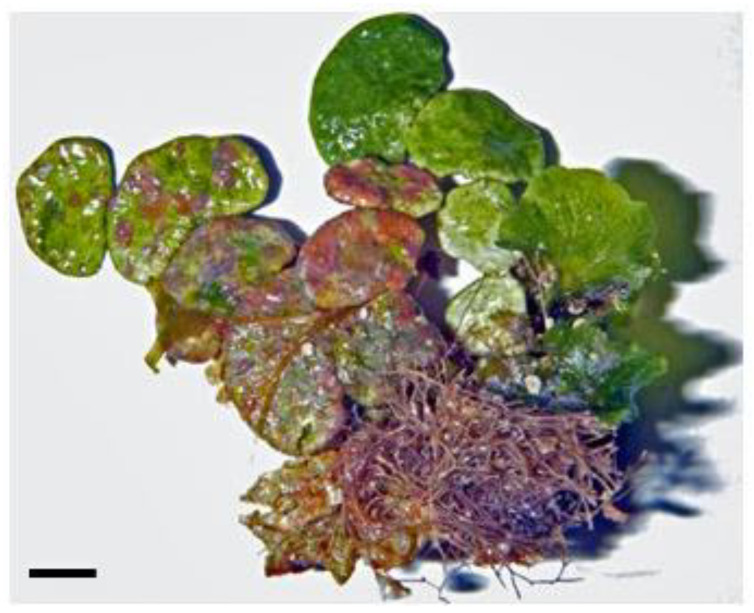
The marine green macroalga, *Halimeda tuna*. Scale = 1 cm.

**Figure 2 life-13-01831-f002:**
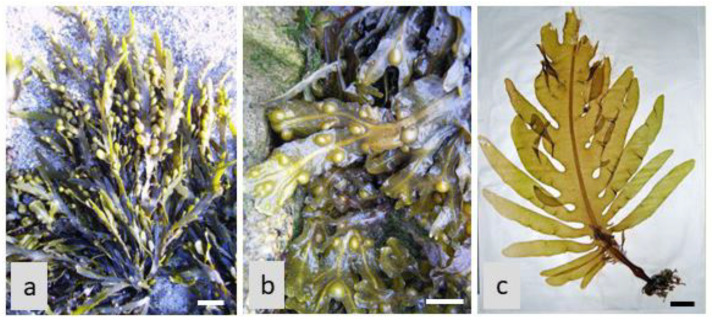
Marine brown macroalgae: (**a**)—*Ascophyllum nodosum*; (**b**)—*Fucus vesiculosus*; (**c**)—*Undaria pinnatifida*. Scale = 1 cm.

**Figure 3 life-13-01831-f003:**
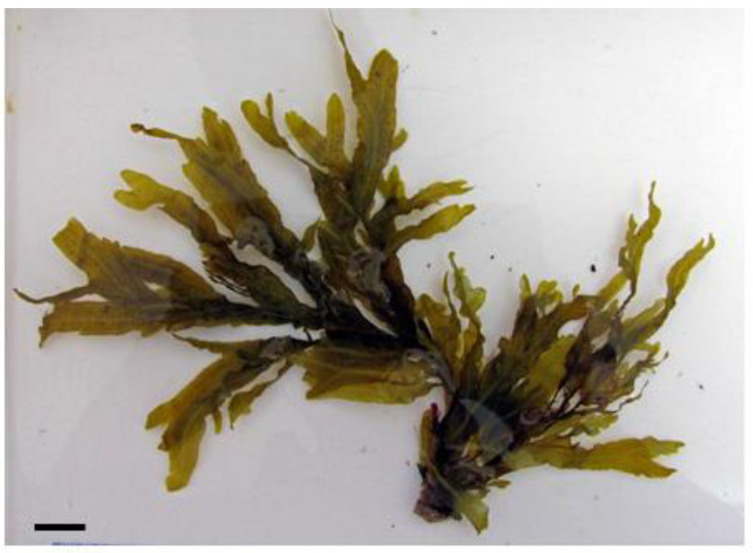
The marine brown macroalga, *Dictyopteris polypodioides* (Phaeophyceae). Scale = 1 cm.

**Figure 4 life-13-01831-f004:**
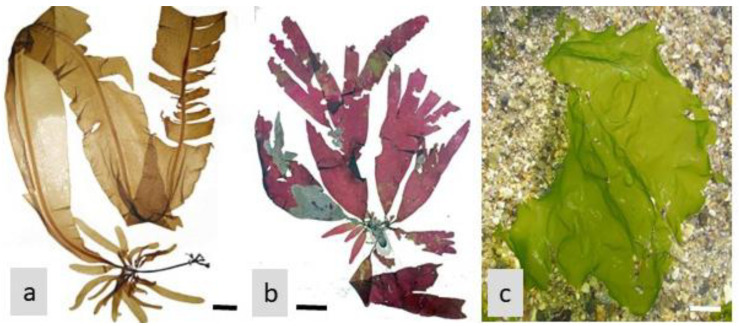
*Alaria* (Phaeophyceae) (**a**); *Palmaria* (Rhodophyta) (**b**); *Ulva* (Chlorophyta) (**c**); Scale = 1 cm.

**Figure 5 life-13-01831-f005:**
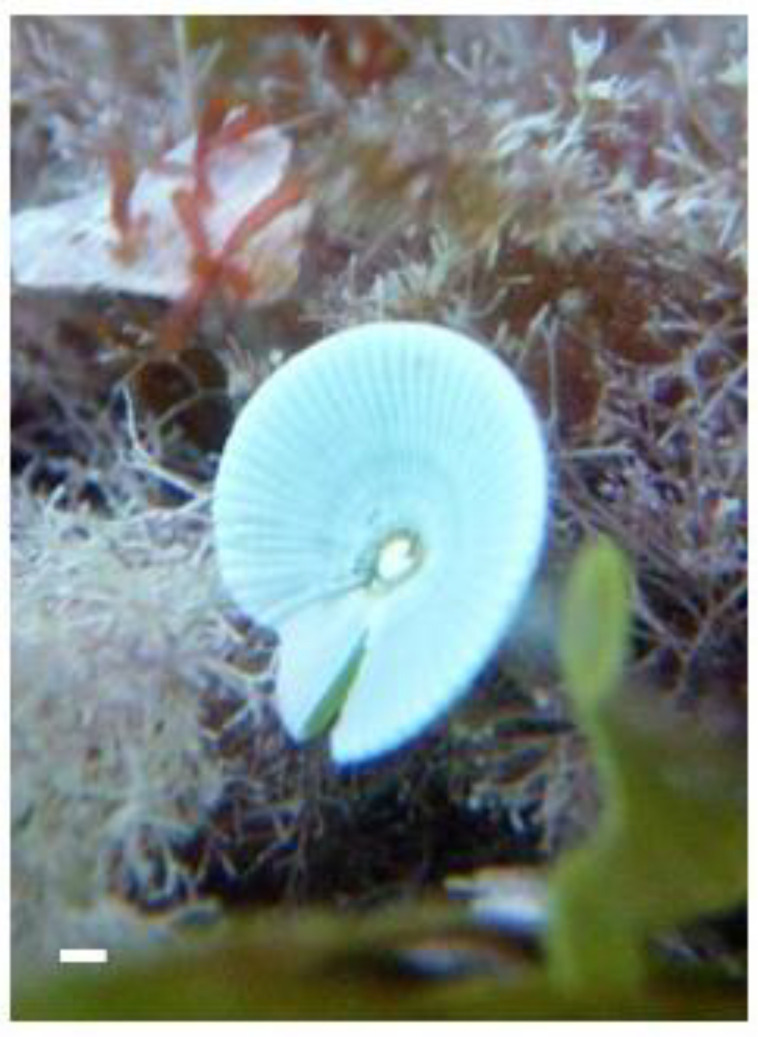
Marine green macroalga: *Acetabularia acetabulum*. Scale = 1 cm.

**Figure 6 life-13-01831-f006:**
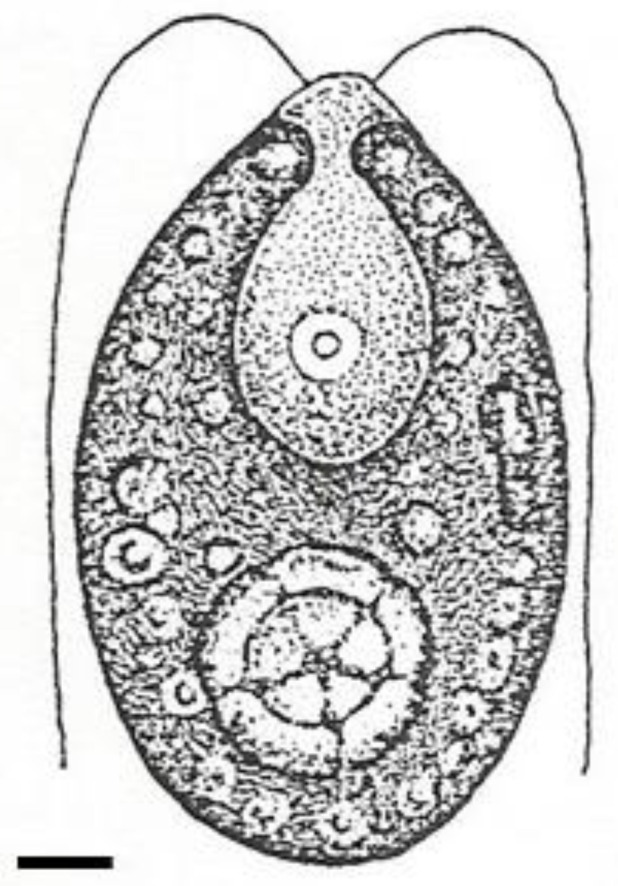
Marine green microalga illustration: *Dunaliella salina*. Scale = 5 µm.

**Figure 7 life-13-01831-f007:**
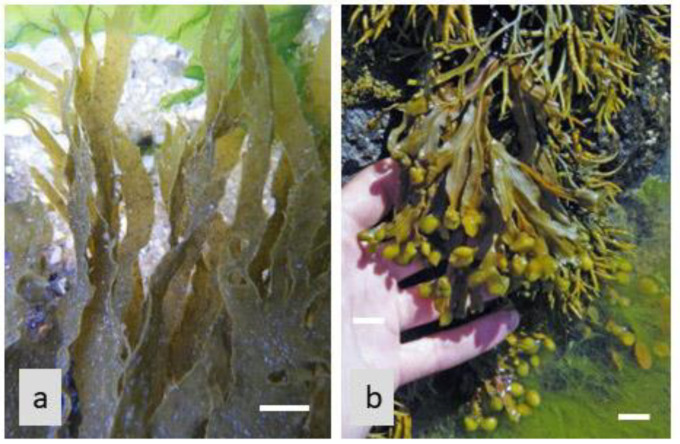
Marine brown macroalgae: (**a**)—*Taonia atomaria*, (**b**)—*Fucus spiralis*. Scale = 1 cm.

**Figure 8 life-13-01831-f008:**
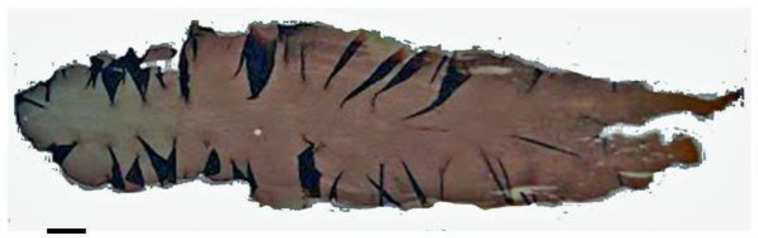
Red macroalga: *Pyropia yezoensis* (Nori). Scale = 1 cm.

**Figure 9 life-13-01831-f009:**
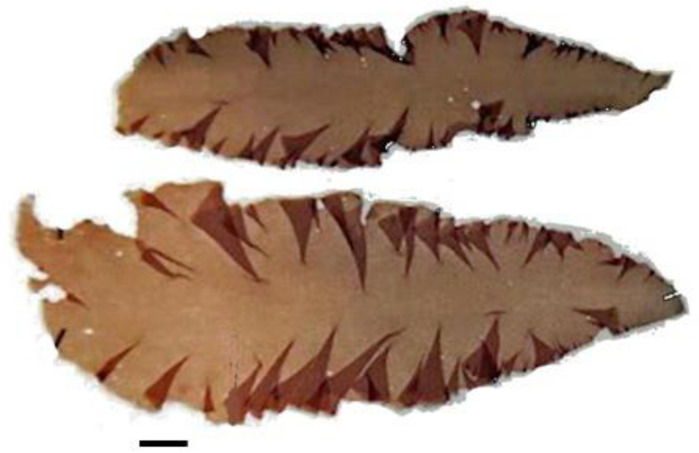
Red macroalga: *Pyropia tenera* (Nori). Scale = 1 cm.

**Figure 10 life-13-01831-f010:**
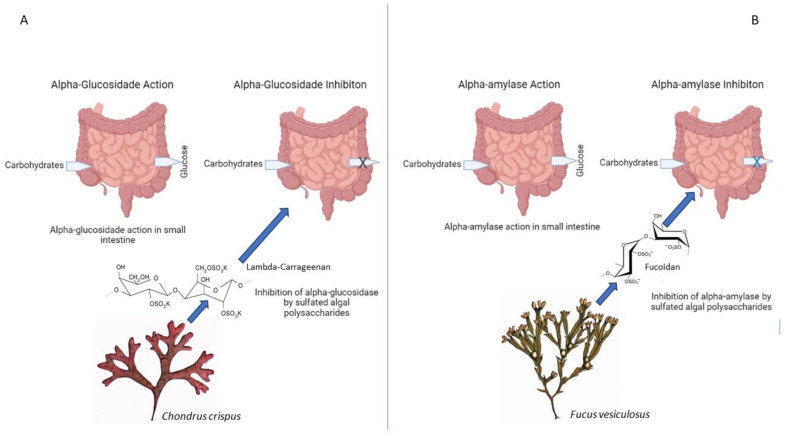
Inhibition of carbohydrate-digesting enzymes by algae-derived compounds. Algae-derived compounds exhibit inhibitory effects on key carbohydrate-digesting enzymes, α-glucosidase (**A**) and α-amylase (**B**). This inhibition results in decreased carbohydrate breakdown, leading to the improved regulation of postprandial glucose levels and reduced hyperglycemia. “X” and “X” = reduction of glucose levels.

**Table 1 life-13-01831-t001:** Glucose levels reduction by algae extracts via miscellaneous mechanisms (adapted from [13,14,15,16,33,44,68,69,70,71,72]).

Species	Extract	Activity	Assay
*Alaria marginata* (P)	Ethyl acetate extract	α-amylase inhibition, IC_50_ = 63.28 μg/mL α-glucosidase inhibition, IC_50_ = 15.66 μg/mL	*In vitro* assay
*Ascophyllum nodosum* (P)	Aqueous ethanolic extract	α-glucosidase inhibition, IC_50_ = 77 μg/mL	*In vitro* assay
*A. nodosum* (P)	Water extract	α-amylase inhibition, IC_50_ = 1.34 μg phenolics α-glucosidase inhibition, IC_50_ = 0.24 μg phenolics	*In vitro* assay
*A. nodosum* (P)	Phlorotannin-rich extract	α-amylase inhibition, IC_50_ ~0.1 μg/mL GAE α-glucosidase inhibition, IC_50_ ~20 μg/mL GAE	*In vitro* assay
*A. nodosum* (P)	Cold water and ethanol extracts	α-amylase inhibition (water), IC_50_ = 53.6 μg/mL α-amylase inhibition (ethanol), IC_50_ = 44.7 μg/mL	*In vitro* assay
*A. nodosum* (P)	Fucoidan	α-amylase inhibition, IC_50_ = 0.12–4.64 mg/mL α-glucosidase inhibition, IC_50_ = 0.013–0.047 mg/mL	*In vitro* assay
*Ulva intestinalis* (C)	Methanolic extract	α-amylase inhibition (500 μg/mL) = 59%	*In vitro* assay
*U. lactuca* (C)	Water extract	α-amylase inhibition, IC_50_ = 67 μg/mL α-glucosidase inhibition, IC_50_ = 53 μg/mL	*In vitro* assay
*U. lactuca* (C)	Methanolic extract	α-amylase inhibition, IC_50_ = 1.00 mg/mL α-glucosidase inhibition, IC_50_ = 1.14 mg/mL	*In vitro* assay
*Ecklonia bicyclis* (P)	Phloroglucinol derivatives	α-amylase inhibition at 1 mM by: Dieckol, inhibition = 97.5% 1-(3′,5′-dihydroxyphenoxy)-7-(2″,4″,6″-trihydroxyphenoxy)-2,4,9-trihydroxydibenzo-1,4-Dioxin, inhibition = 89.5% Eckol, inhibition = 87.5%	*In vitro* assay
*E. bicyclis* (P)	Phlorofucofuroeckol-A Dieckol 7-Phloroeckol Eckol Dioxinodehydroeckol Phloroglucinol	α-glucosidase inhibition by: Phlorofucofuroeckol-A, IC_50_ = 1.37 μM Dieckol, IC_50_ = 1.61 μM 7-Phloroeckol, IC_50_ = 6.13 μM Eckol, IC_50_ = 22.78 μM Dioxinodehydroeckol, IC_50_ = 34.60 μM Phloroglucinol, IC_50_ = 141.18 μM	*In vitro* assay
*E. bicyclis* (P)	Fucofuroeckol A	α-amylase inhibition, IC_50_ = 42.91 μM α-glucosidase inhibition, IC_50_ = 131.34 nM	*In vitro* assay
	Dioxinodehydroeckol	α-amylase inhibition, IC_50_ = 472.70 μM α-glucosidase inhibition, IC_50_ = 93.33 nM	
*E. cava* (P)	Dieckol Fucodiphloroethol G Phlorofucofuroeckol A 6,6′-bieckol 7-Phloroeckol	α-glucosidase inhibition by: Dieckol, IC_50_ = 10.8 μM Fucodiphloroethol G, IC_50_ = 19.5 μM Phlorofucofuroeckol A, IC_50_ = 19.7 μM 6,6′-bieckol, IC_50_ = 22.2 μM 7-Phloroeckol, IC_50_ = 49.5 μM	*In vitro* assay
	Dieckol 7-Phloroeckol 6,6′-bieckol Fucodiphloroethol G Phlorofucofuroeckol A	α-amylase inhibition by: Dieckol, IC_50_ = 124.9 μM 7-Phloroeckol, IC_50_ = 250.0 μM 6,6′-bieckol, IC_50_ > 500 μM Fucodiphloroethol G, IC_50_ > 500 μM Phlorofucofuroeckol A, IC_50_ > 500 μM	
*E. cava* (P)	Dieckol	α-amylase inhibition, IC_50_ = 0.66 mM α-glucosidase inhibition, IC_50_ = 0.24 mM	*In vitro* assay
*E. maxima* (P)	Eckol Dibenzo [1,4] dioxine-2,4,7,9-tetraol Phloroglucinol	α-glucosidase inhibition by: Eckol, IC_50_ = 11.16 μM Dibenzo [1,4] dioxine-2,4,7,9-tetraol, IC_50_ = 33.69 μM Phloroglucinol, IC_50_ = 1991 μM	*In vitro* assay
*Ecklonia cava* subsp. *stolonifera* (formerly *E. stolonifera*) (P)	Water extract	α-glucosidase inhibition against: α-glucosidase (*Saccharomyces*—Yeast), IC_50_ = 0.026 mg/mL Rat intestinal maltase, IC_50_ = 4.213 mg/mL Rat intestinal sucrose, IC_50_ = 10.10 mg/mL Rat intestinal isomaltase, IC_50_ > 100 mg/mL Rat intestinal glucoamylase, IC_50_ > 100 mg/mL	*In vitro* assay
	Methanolic extract	α-glucosidase inhibition against: α-glucosidase (Saccharomyces), IC_50_ = 0.022 mg/mL Rat intestinal maltase, IC_50_ = 0.772 mg/mL Rat intestinal sucrose, IC_50_ = 4.056 mg/mL Rat intestinal isomaltase, IC _50_ > 100 mg/mL Rat intestinal glucoamylase, IC_50_ = 5.851 mg/mL	
*E. cava* subsp. *stolonifera* (P)	Phlorofucofuroeckol-A Dieckol 7-Phloroeckol Eckol Dioxinodehydroeckol Phloroglucinol	α-glucosidase inhibition by: Phlorofucofuroeckol-A, IC_50_ = 1.37 μM Dieckol, IC_50_ = 1.61 μM 7-Phloroeckol, IC_50_ = 6.13 μM Eckol, IC_50_ = 22.78 μM Dioxinodehydroeckol, IC_50_ = 34.60 μM Phloroglucinol, IC_50_ = 141.18 μM	*In vitro* assay
*Eucheuma denticulatum* (R)	Etanholic extract	α-amylase inhibition (10 mg/mL) = 67%	*In vitro* assay
*Fucus distichus* (P)	Ethyl acetate extract	α-amylase inhibition, IC_50_ = 13.98 μg/mL α-glucosidase inhibition, IC_50_ = 0.89 μg/mL	*In vitro* assay
*F. vesiculosus* (P)	Cold water and ethanol extracts	α-glucosidase inhibition (water), IC_50_ = 0.32 μg/mL α-glucosidase inhibition (ethanol), IC_50_ = 0.49 μg/mL	*In vitro* assay
*F. vesiculosus* (P)	Fucoidan	α-glucosidase inhibition, IC_50_ = 0.049 mg/mL	*In vitro* assay
*Gracilaria corticata* (R)	Water extract	α-amylase inhibition, IC_50_ = 82 µg/mL α-glucosidase inhibition, IC_50_ = 87 µg/mL	*In vitro* assay
*G. edulis* (R)	Water extract	α-amylase inhibition, IC_50_ = 83 µg/mL α-glucosidase inhibition, IC_50_ = 46 µg/mL	*In vitro* assay
*G. edulis* (R)	Methanolic extract	α-amylase inhibition (300 μg/mL) = 57%	*In vitro* assay
	Acetone extract	α-amylase inhibition (300 μg/mL) = 53%	
	Acetate extract	α-amylase inhibition (300 μg/mL) = 60%	
*G. gacilis* (R)	Methanolic extract	α-amylase inhibition (300 μg/mL) = 68%	*In vitro* assay
*G. opuntia* (R)	Water extract (sulfated polygalactans)	α-amylase inhibition, IC_50_ = 0.04 mg/mL α-glucosidase inhibition, IC_50_ = 0.09 mg/mL	*In vitro* assay
*Grateloupia elliptica* (R)	2,4,6-Tribromophenol	α-glucosidase inhibition against: *B. stearothermophilus*, IC_50_ = 130.3 μM *S. cerevisiae*, IC_50_ = 60.3 μM Rat intestinal maltase, IC_50_ = 5.0 mM Rat intestinal sucrose, IC_50_ = 4.2 mM	*In vitro* assay
	2,4-Dibromophenol	α-glucosidase inhibition against: *Bacillus stearothermophilus*, IC_50_ = 230.3 μM *S. cerevisiae*, IC_50_ = 110.4 μM Rat intestinal maltase, IC_50_ = 4.8 mM Rat intestinal sucrose, IC_50_ = 3.6 mM	
*Kappaphycus alvarezii* (R)	Water extract (sulfated polygalactans)	α-amylase inhibition, IC_50_ = 0.15 mg/mL α-glucosidase inhibition, IC_50_ = 0.09 mg/mL	*In vitro* assay
*Odonthalia corymbifera* (R)	Bromophenols	α-glucosidase (*S. cerevisiae*) inhibition by: 1,4-Butanediol diglycidyl ether, IC_50_ = 0.098 μM 4-Bromo-2,3-dihydroxy-6-hydroxymethylphenyl 2,5- dibromo-6-hydroxy-3-hydroxymethylphenyl ether, IC_50_ = 25.0 μM 4-Bromo-2,3-dihydroxy-6-methoxymethylphenyl 2,5-dibromo-6-hydroxy-3-methoxymethylphenyl ether, IC_50_ = 53.0 μM 2,3-dibromo-4,5-dihydroxybenzyl alcohol, IC_50_ = 89.0 μM	*In vitro* assay
*Polyopes lancifolius* (R)	1,4-Butanediol diglycidyl ether	α-glucosidases inhibition against: *B. stearothermophilus*, IC_50_ = 0.12 μM *S. cerevisiae*, IC_50_ = 0.098 μM Rat intestinal maltase, IC_50_ = 1.20 mM Rat intestinal sucrose, IC_50_ = 1.00 mM	*In vitro* assay
*Portieria hornemannii* (as *Chondrococcus hornemannii*) (R)	Methanolic extract	α-amylase inhibition (300 μg/mL) = 61%	*In vitro* assay
	Acetate extract	α-amylase inhibition (300 μg/mL) = 94%	
*Sargassum patens* (P)	2-(4-(3,5 (dihydroxyphenoxy)-3,5-dihydroxyphenoxy) benzene-1,3,5- triol	α-amylase inhibition, IC_50_ = 3.2 μg/mL α-glucosidase inhibition against: Rat intestinal maltase, IC_50_ = 114.0 μg/mL Rat intestinal sucrose, IC_50_ = 25.4 μg/mL	*In vitro* assay
*S. polycystum* (P)	Water extract	α-amylase inhibition, IC_50_ = 60 µg/mL α-glucosidase inhibition, IC_50_ = 50 µg/mL	*In vitro* assay
*S. polycystum* (P)	Ethanolic extract	α-amylase inhibition (10 mg/mL) = 46%	*In vitro* assay
*S. ringgoldianum* (P)	Methanolic extract	α-amylase inhibition, IC_50_ = 0.18 mg/mL α-glucosidase inhibition, IC_50_ = 0.12 mg/mL	*In vitro* assay
*S. swartzii* (as *S. wightii*) (P)	Methanolic extract	α-amylase inhibition (300 μg mL^−1^) = 53%	*In vitro* assay
*S. swartzii* (as *S. wightii*) (P)	Fucoidan	α-glucosidase inhibition, IC_50_ = 132 µg	*In vitro* assay
*S. swartzii* (as *S. wightii*) (P)	Ethanol extraction and CaCl_2_ precipitation method (Fucoidan)	α-amylase inhibition, IC_50_ = 103.83 μg/mL	*In vitro* assay
*S. tenerrimum* (P)	Methanolic extract	α-amylase inhibition (300 μg/mL) = 65%	*In vitro* assay
*Spatoglossum asperum* (P)	Methanolic extract	α-amylase inhibition, IC_50_ = 55 µg/mL α-glucosidase inhibition, IC_50_ = 61 µg/mL	*In vitro* assay
*Spatoglossum schroederi* (P)	Acetone crude extract	α-amylase inhibition, IC_50_ = 0.58 mg/mL	*In vitro* assay
*Symphyocladia latiuscula* (P)	Bromophenols	2,3,6-Tribromo-4,5-dihydroxybenzyl Alcohol, IC_50_ = 11.0 μM Sucrose inhibition, IC_50_ = 2.4	*In vitro* assay
*Turbinaria ornata* (P)	Methanolic extract	α-amylase inhibition (300 μg/mL) = 45%	*In vitro* assay

C—Chlorophyta; R—Rhodophyta; P—Phaeophyceae.

## Data Availability

Not applicable.

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
