# Peer review of "Algae-Derived Natural Products in Diabetes and Its Complications—Current Advances and Future Prospects"

_life, 2023, doi:10.3390/life13091831_

Round 1

Reviewer 1 Report

The manuscript contains valuable and recent information related to diabetes. 

Although the authors make a comment, it is of fundamental  to emphasize the importance of toxicological studies. The general public may be left with the impression that there is no danger to human health using of algae without knowing the secondary metabolites and effects they may have.  

Author Response

Reviewer 1

The manuscript contains valuable and recent information related to diabetes.

Although the authors make a comment, it is of fundamental to emphasize the importance of toxicological studies. The general public may be left with the impression that there is no danger to human health using of algae without knowing the secondary metabolites and effects they may have. 

Answer: The authors are grateful for the careful review and for the suggested suggestions, which they accept.

The reviewer comment highlights a critical aspect that the authors should address regarding their discussion on algae-derived products. The reviewer points out that while the authors do acknowledge the importance of safety assessments and toxicity studies, it is fundamental to emphasize the significance of toxicological studies more explicitly. By doing so, the authors can ensure that the general public does not form a misconception that algae-based products are entirely devoid of potential risks to human health. The reviewer's concern stems from the fact that such products might contain secondary metabolites with unforeseen effects, and failing to communicate this possibility could lead to a false sense of security among consumers. In light of this, it's essential for the authors to further underscore the need for rigorous toxicological investigations to comprehensively understand the potential impacts of these products on human health. By addressing this concern, the authors can provide a more balanced and accurate perspective, helping individuals make informed decisions about the use of algae-derived products.

In this sense, this paragraph to “6.1.3. Safety and Toxicity Assessment” were added:

“Recent advances in the field of natural products derived from algae have brought about a growing awareness of their potential benefits. However, it is critical to pay attention to the importance of thorough toxicological studies [13]. While algae-derived products are generally considered safe, contemporary research underscores the need for a nuanced understanding of their secondary metabolites and potential effects on human health [34]. Several studies have highlighted the intricate bioactive compounds present in algae, some of which might have yet-to-be-discovered physiological or toxicological implications [32]. Neglecting to communicate the potential risks associated with these secondary metabolites could inadvertently mislead the public into believing that algae-based interventions are entirely risk-free [6]. Therefore, current discourse emphasizes the necessity of comprehensive toxicological investigations to uncover any hidden adverse effects and to establish accurate risk assessments. By openly addressing these concerns, both researchers and the general public can make informed decisions regarding the utilization of algae-derived products, ensuring not only their efficacy but also their safety for human health [16].”

Reviewer 2 Report

The content of this review paper is useful, gathering together studies on aspects of algal extracts and their potential in addressing diabetes.  the images of seaweeds were useful to the reader. 

However, the structure of writing  is both unusual and a little distracting. there is a lot of repetition and 'padding' in 1.1, 1.2.  Condensing these into one would make sense.

The review has a "directory like'feel , pointing the reader to other papers, but without much direction.  

 Please consider regrouping the nest short sub- paragraphs into larger subsections, and avoid repetition.  Tables would be easier to read perhaps? One of the aspects missing here is any quantitation of effect, or opinion from the author.   For example, there is no information on any of the bioavailability studies that the author points to in 6.1.2. 

The section on clinical trials 6.2.6 does not list any clinical trial papers on algal products- and should be combined with 6.2.3

Graphical images would perhaps enhance this manuscript.

The abstract would be improved by briefly articulating the algal products with the most potential, and which aspects of diabetes would be addressed.

My comments here relate not to spelling (which is perfect) but the underlying structure of the writing.

There is repetition and unnecessary padding at the beginning- and then so many short paragraphs.  The reader is left wanting to know what the relative effects are- which extracts look the most promising ?

Are any of these compounds in food, or available in the market to date either as drugs, medical devices?

Author Response

Reviewer 2

The content of this review paper is useful, gathering together studies on aspects of algal extracts and their potential in addressing diabetes.  the images of seaweeds were useful to the reader.

However, the structure of writing is both unusual and a little distracting. there is a lot of repetition and 'padding' in 1.1, 1.2.  Condensing these into one would make sense.

Answer: The authors are grateful for the careful review and for the suggested suggestions, which they accept.

Some of the paragraphs have been reduced and condensed, as suggested.

The authors warn, however, that it was not always possible to reduce the subparagraphs. The titles of these are fundamental for the understanding and consultation of the different species of algae, their different compounds and their bioactivities.

The review has a "directory like’ feel, pointing the reader to other papers, but without much direction. 

Answer: Yes, the purpose of this review is to serve as a directory, so that readers have a base to find research articles on each of the topics and subjects reviewed in this work more easily.

 Please consider regrouping the nest short sub- paragraphs into larger subsections, and avoid repetition.  Tables would be easier to read perhaps? One of the aspects missing here is any quantitation of effect, or opinion from the author.   For example, there is no information on any of the bioavailability studies that the author points to in 6.1.2.

Answer: The authors reorganized some of the parts of this review, reducing the number of paragraphs and merging some parts of the text into single paragraphs, as suggested.

The section on clinical trials 6.2.6 does not list any clinical trial papers on algal products- and should be combined with 6.2.3

Answer: Some restructuring of these paragraphs has been done, as suggested.

Graphical images would perhaps enhance this manuscript.

Answer: (5.2.5. Alpha-glucosidase and alpha-amylase) Added Figure 10. Inhibition of carbohydrate-digesting enzymes by algae-derived compounds. Algae-derived compounds exhibit inhibitory effects on key carbohydrate-digesting enzymes, α-glucosidase and α-amylase. This inhibition results in decreased carbohydrate breakdown, leading to improved regulation of postprandial glucose levels and reduced hyperglycemia.

Note: This figure can also serve as the “Graphical Abstract” of the manuscript.

The abstract would be improved by briefly articulating the algal products with the most potential, and which aspects of diabetes would be addressed.

Answer: The abstract was improved: “Diabetes poses a significant global health challenge, necessitating innovative therapeutic strategies. Natural products and their derivatives have emerged as promising candidates for diabetes management due to their diverse compositions and pharmacological effects. Algae, in particular, have garnered attention for their potential as a source of bioactive compounds with antidiabetic properties. This review offers a comprehensive overview of algae-derived natural products for diabetes management, highlighting recent developments and future prospects. It underscores the pivotal role of natural products in diabetes care, delves into the diversity of algae, their bioactive constituents, and underlying mechanisms of efficacy. Noteworthy algal derivatives with substantial potential are briefly elucidated, along with their specific contributions to addressing distinct aspects of diabetes. The challenges and limitations inherent in utilizing algae for therapeutic interventions are examined, accompanied by strategic recommendations for optimizing their effectiveness. By ad-dressing these considerations, this review aims to chart a course for future research in refining algae-based approaches. Leveraging the multifaceted pharmacological activities and chemical com-ponents of algae holds significant promise in the pursuit of novel antidiabetic treatments. Through continued research and fine-tuning of algae-based interventions, the global diabetes burden could be mitigated, ultimately leading to enhanced patient outcomes.”

Comments on the Quality of English Language

My comments here relate not to spelling (which is perfect) but the underlying structure of the writing.

Answer: Some restructuring of these paragraphs has been done, as suggested.

There is repetition and unnecessary padding at the beginning- and then so many short paragraphs. 

Answer: Some restructuring of these paragraphs has been done, as suggested.

The reader is left wanting to know what the relative effects are- which extracts look the most promising ?

Answer: 3 paragraphs and the Table 1 were added to 2.3: The effectiveness of algae-derived natural products in diabetes management stems from their diverse array of mechanisms of action. These bioactive constituents exhibit the capacity to intervene in crucial pathways associated with glucose homeostasis, insulin sensitivity, and the prevention or alleviation of diabetes-related complications. A comprehensive understanding of these mechanisms is pivotal in harnessing the full therapeutic potential of algae-derived natural products [33,67].

While certain algal derivatives showcase considerable potential in ameliorating anti-diabetic effects, a rigorous comparison of their relative efficacy remains a subject demanding further exploration. Algae extracts exhibit variegated profiles of bioactive components, each potentially addressing distinct aspects of diabetes. Some extracts may exhibit superior effects on glucose regulation, while others could excel in enhancing insulin sensitivity or mitigating complications (Table 1). Identifying the most promising algae extracts necessitates meticulous investigation, encompassing both in vitro and in vivo studies, along with comparative assessments of their therapeutic impact [13,44].

Prominent candidates may emerge from algae species that have consistently demonstrated potent antidiabetic effects across multiple studies. Algae extracts rich in specific bioactive compounds, such as polyphenols, polysaccharides, or peptides, could hold the key to enhanced antidiabetic efficacy. Nonetheless, ascertaining the hierarchy of promising extracts demands rigorous scientific scrutiny, including rigorous clinical trials to corroborate their effects in human populations. This collective pursuit not only refines our understanding of algae's therapeutic potential but also paves the way for optimized interventions that can tangibly alleviate the burden of diabetes and improve patient outcomes [32,33].

Are any of these compounds in food, or available in the market to date either as drugs, medical devices?

Answer: The following paragraph has been added:

“7.1 Several compounds extracted from seaweed utilized in food, drugs, or medical devices.

Several compounds extracted from seaweed are utilized in food, drugs, or medical devices. Alginate (E400-E405) is a polysaccharide extracted from brown seaweeds. It's commonly used as a food additive, particularly in the form of sodium alginate, for its gel-ling and thickening properties. It's used to create gels, encapsulate flavors, and modify the texture of foods. Some seaweed-derived materials are used in medical applications. Algi-nate dressings, for instance, are used to treat wounds due to their absorbent and hemo-static properties [168].

Carrageenan (E407) is another polysaccharide derived from red seaweeds. It's used as a food stabilizer, thickener, and gelling agent in products like dairy, meat, and processed foods. It's also used in pharmaceuticals as an inactive ingredient in tablet formulations. Carrageenan has been explored for its potential in drug delivery systems due to its bio-compatibility and ability to form gels. It can be used to encapsulate drugs and provide controlled release [169].

Fucoidan is a complex sulfated polysaccharide found in brown seaweeds. It has shown potential health benefits, including immune system modulation and potential an-ti-inflammatory properties. While not yet approved as a drug, fucoidan supplements are available in some markets for their potential health-promoting effects [170].

Codium (Chlorophyta) has been investigated for its potential as an antidiabetic agent. Some studies have shown that Codium extracts may have glucose-lowering effects and could be beneficial in managing diabetes. Seaweeds are also rich sources of iodine, a vital mineral for thyroid function. Iodine extracted from seaweeds can be used in iodized salt and dietary supplements to address iodine deficiency [171].

Agar is a gelatinous substance derived from red seaweeds. It's used in the food in-dustry as a vegetarian alternative to gelatin in products like desserts, gummy candies, and microbiological culture media [172].”

Reviewer 3 Report

Highly recommend the use of a graphical abstract as this article covers a wide range of algal bioactive compounds against diabetics.

Would be appropriate if you combine parts 2, 3, 4, and 5 to generate a much bigger picture where relevant. 

Each of the above parts describes a limited amount of information which is not expected from a review article. For instance, polysaccharides are sub-sectioned as a bioactive compound with antidiabetic potential. But authors do not try to touch the depths associated with it the related activity. It would be beneficial if the authors discussed the chemistry, biochemistry, and SAR related to each bioactive compound with illustrations included rather than briefly sectioning them. 

Due to the lack of proper in-depth analysis and comprehensive discussions, I recommend the rejection of this manuscript.

NA

Author Response

Reviewer 3

Highly recommend the use of a graphical abstract as this article covers a wide range of algal bioactive compounds against diabetics.

Answer: The authors are grateful for the comments made so, together with the corrections made in agreement with you and the other two reviewers, which we thank, we have added Figure 10, which may also serve as a Graphical Abstract

Would be appropriate if you combine parts 2, 3, 4, and 5 to generate a much bigger picture where relevant.

Answer: The corrections are made in agreement with you and the other two reviewers

Each of the above parts describes a limited amount of information which is not expected from a review article. For instance, polysaccharides are sub-sectioned as a bioactive compound with antidiabetic potential. But authors do not try to touch the depths associated with it the related activity. It would be beneficial if the authors discussed the chemistry, biochemistry, and SAR related to each bioactive compound with illustrations included rather than briefly sectioning them.

Answer: New paragraphs and Figure 10 were added, in order to elucidate in a synthetic way some of the associated mechanisms.

Due to the lack of proper in-depth analysis and comprehensive discussions, I recommend the rejection of this manuscript.

Answer: The authors hope that with the corrections made, the manuscript is more robust and that it satisfies all reviewers. Thank you for your time and comments!

Round 2

Reviewer 3 Report

The authors have not sufficiently responded to the improvement of the manuscript. Adding a table to roughly demonstrate the ideas and combining subsections of part 6 does not reflect the proper significant revisioning of their manuscript. Point-by-point responses have not been made sufficiently. Overall, the authors have not provided commitment toward revision.

NA

Author Response

Reviewer 3

Highly recommend the use of a graphical abstract as this article covers a wide range of algal bioactive compounds against diabetics.

Answer: As requested, a Graphical Abstract was added.

Would be appropriate if you combine parts 2, 3, 4, and 5 to generate a much bigger picture where relevant.

Answer: With all due respect, allow me to disagree with your proposal. This review was designed and structured with these 4 separate parts, so it would not make sense to merge them into a single part.

Part 2 corresponds to the presentation of seaweeds and their main characteristics. The 3rd part corresponds to compounds with bioactivities in glycemic control, the 4th part talks about compounds from algae with potential in controlling diabetes, the 5th part corresponds to Mechanistic Insights and Molecular Targets. Merging these 4 parts would mischaracterize the ideal basis of this review.

Each of the above parts describes a limited amount of information which is not expected from a review article. For instance, polysaccharides are sub-sectioned as a bioactive compound with antidiabetic potential. But authors do not try to touch the depths associated with it the related activity. It would be beneficial if the authors discussed the chemistry, biochemistry, and SAR related to each bioactive compound with illustrations included rather than briefly sectioning them.

Answer: New paragraphs and Figure 10 were added, in order to elucidate in a synthetic way some of the associated mechanisms.